# The Alter Retina: Alternative Splicing of Retinal Genes in Health and Disease

**DOI:** 10.3390/ijms22041855

**Published:** 2021-02-12

**Authors:** Izarbe Aísa-Marín, Rocío García-Arroyo, Serena Mirra, Gemma Marfany

**Affiliations:** 1Departament of Genetics, Microbiology and Statistics, Avda. Diagonal 643, Universitat de Barcelona, 08028 Barcelona, Spain; iaisa@ub.edu (I.A.-M.); rociogarciaarroyo@ub.edu (R.G.-A.); serena.mirra@ub.edu (S.M.); 2Centro de Investigación Biomédica en Red Enfermedades Raras (CIBERER), Instituto de Salud Carlos III (ISCIII), Universitat de Barcelona, 08028 Barcelona, Spain; 3Institute of Biomedicine (IBUB, IBUB-IRSJD), Universitat de Barcelona, 08028 Barcelona, Spain

**Keywords:** retina, alternative splicing, inherited retinal dystrophies, splicing factors, non-canonical splice site variants, deep intronic variants, microexons

## Abstract

Alternative splicing of mRNA is an essential mechanism to regulate and increase the diversity of the transcriptome and proteome. Alternative splicing frequently occurs in a tissue- or time-specific manner, contributing to differential gene expression between cell types during development. Neural tissues present extremely complex splicing programs and display the highest number of alternative splicing events. As an extension of the central nervous system, the retina constitutes an excellent system to illustrate the high diversity of neural transcripts. The retina expresses retinal specific splicing factors and produces a large number of alternative transcripts, including exclusive tissue-specific exons, which require an exquisite regulation. In fact, a current challenge in the genetic diagnosis of inherited retinal diseases stems from the lack of information regarding alternative splicing of retinal genes, as a considerable percentage of mutations alter splicing or the relative production of alternative transcripts. Modulation of alternative splicing in the retina is also instrumental in the design of novel therapeutic approaches for retinal dystrophies, since it enables precision medicine for specific mutations.

## 1. Introduction

The human genome is estimated to contain ∼30,000 genes, which represent about 1% of the total genome [1,2]. Although the number of genes is not much higher than in invertebrates, a higher molecular complexity can be attained by the production of multiple mRNA isoforms from a single gene, generated by alternative splicing (AS) and leading to a huge diversification of the proteome [3]. Alternative splicing events (ASEs) of pre-mRNA transcripts strongly contribute to the cellular regulatory landscape, protein diversity, and therefore, to organismal complexity, together with other related mechanisms such as the use of alternative promoters, transcription start sites and polyadenylation sites. 

AS consists in the selective removal or retention of introns and exons giving rise to different rearranged patterns of mature mRNA products. In the classical view, AS may occur via different patterns of binary events involving two exons or two splice sites in the same exon. Binary AS includes exon skipping, mutually exclusive exons, alternative donor (5′) and receptor (3′) splice sites, and intron retention. However, this vision does not take account of complex splicing events [4]. The proper recognition of pre-mRNA regions during splicing is mediated by cis-acting regulatory elements, whose location on the pre-mRNA spatially orchestrates the splicing event. Cis splicing elements include the splice donor site (SDS), the splice acceptor site (SAS), the branch-site (BS) and the polypyrimidine tract (Py) upstream of the 3′ splice site. In addition, cis-regulatory sequences can interact with trans-acting regulatory elements to promote or repress splicing [5]. Splicing activators include serine- and arginine-rich (SR) proteins, while splicing repressors include hnRNPs [6,7]. A number of other splicing activators or repressors have been identified, and the combinatory binding of these elements in a time- and space-specific manner originates high specific isoform patterns characteristic of each cell-type. Moreover, chromatin organization can also change or determine splicing patterns. The spliceosome catalyzes splicing in a two-step trans-esterification reaction that joins each donor site to the correspondent acceptor site [8]. 

Deep transcriptome sequencing of the human genome showed that the frequency of AS increases with species complexity, with almost all multi-exon human genes being alternatively spliced in humans [9,10]. AS occurs in all tissues and is subject to cell-specific and developmental-specific regulation [11]. Moreover, recent advances in genomic technologies and computational tools employed in genome-wide studies allowed us to shed light on the wide range of alternative splice isoforms generated in the context of homeostatic adaptation and diseases [12]. 

The most common occurrence of ASEs is observed in the central nervous system (CNS) of vertebrates [13], where it plays pivotal roles in processes such as neurogenesis, cell migration, synaptogenesis, synaptic function or neuronal network function and plasticity [14]. Remarkably, the CNS needs to rapidly adapt its physiology to dynamic environmental changes. Tissue- and cell-specific modulation of the AS patterns provides a brilliant solution to this high demand of plastic adaptation, guarantying an efficient adjustment of network dynamics. Indeed, it allows to produce several mRNA isoforms with specific regulatory features, such as stability or translational efficiency. Additionally, the set of protein isoforms arising from ASEs can vary with respect to their subcellular localization, protein or metabolite interactome and functional features.

The retina, the light sensing tissue of the eye, is a highly specialized tissue belonging to the CNS. Remarkably, high levels of ASEs occur in the retina where precise gene regulation is required for neuronal development and function. The importance of alternative splicing in retina is highlighted by numerous examples where splicing alterations underly retinal disorders and disease, such as cone-rod dystrophy, Usher syndrome (USH) and retinitis pigmentosa (RP). Notably, splicing mutations account for 9% of all disease-causing mutations reported in the Human Gene Mutation Database (HGMD) [15,16]. They include both mutations in gene sequence (exons or introns) compromising the precision of intron removal during the AS (e.g. generating a reading frame shift resulting in a dysfunctional protein) and mutations in proteins directly involved in splicing mechanisms. Then, they can be grouped in cis-acting mutations and trans-acting mutations respectively. Of note, cis-acting mutations located in the exons may impair the splicing pattern without altering the coding sequence [17]. These mutations are hard to identify and are often miscategorized as missense, nonsense, or silent, although they should be considered as splicing mutations.

In this review we will focus on the retinal specific mechanisms leading to the exquisitely regulated execution of the alternative splicing program in the mammalian retina. Moreover, we will pay attention on one of the most challenging aspects of the inherited retinal genetic diagnosis: the identification of mutations altering splicing process or the production of alternative splicing products. Finally, we will summarize the current therapeutic approaches to modulate splicing events in retinal disorders.

## 2. Alternative Splicing of Retinal Genes

### 2.1. Alternative Splicing in the Retina

In the retina, AS represents a crucial regulatory step of gene expression during development and homeostasis. Over 95% of human multiexon genes undergo AS, resulting in mRNA splice variants that are variably expressed between different cell and tissue types [9,18]. Many different types of AS have been identified in the retina, including cassette exons [19], exon skipping [20] and intron retention events [21], mutually exclusive exons [22], alternative splice sites [23,24], alternative promoters [25,26] and alternative polyadenylation (polyA) sites [27] (Figure 1A). 

Mechanisms generating different transcripts from a single locus serve to diversify mRNA sequences, and allow to display a range of protein isoforms that often differ in their function. Interestingly, a specialized type of splicing leading to incorporation of alternative microexons (exons that are ≤30 nt) has been shown to impact neuronal differentiation and function [28]. Even though they represent only 1% of all AS, microexons are the most highly conserved component of developmental alternative splicing regulation. Furthermore, they are enriched for lengths that are multiple of 3 nucleotides and are thus likely to produce alternative protein isoforms [29]. Microexons impact on specific protein regulatory domains, are associated with late neurogenesis and appear altered in neurological disorders [29,30,31,32,33] (Figure 1B).

Some splicing errors can cause frameshift and premature protein truncation, thus resulting in transcripts that are recognized by the cellular mRNA control machinery and are degraded by nonsense-mediated decay (NMD). Therefore, NMD can serve both as a mechanism to target non-functional mRNAs as well as to fine-tune gene expression by regulating the abundance of multiple transcript isoforms from a single gene locus [34].

Some of these ASEs are particularly important in the retina. More than 7000 cassette exons (included only in some transcripts, Figure 1A) that showed differential AS patterns associated with distinct cell types and developmental stages have been reported [19]. Many of the cassette exons belong to retina-specific genes required for homeostasis maintenance in the adult retina, thus indicating that adult genes produce unique isoforms at certain stages, which may play a role in the differentiation maintenance rather than in development. Stage-specific alternative polyA sites have also been reported during retina development [27]. Alternative polyA generates transcript variants that may present different coding regions or 3′ untranslated regions (UTRs) (Figure 1A), whose variability can affect stability, localization, transport and translational properties of the mRNA. Authors found embryonic-specific polyA sites associated with genes involved in cell cycle and cytoskeleton, consistent with the high abundance of dividing progenitor cells. On the contrary, during the early postnatal period (when photoreceptors differentiate and mature), polyA sites were associated with genes involved in phototransduction and visual function [27]. These findings highlight the precise temporal and spatial regulation of AS and its importance controlling gene expression in the retina.

### 2.2. Novel Approaches to Detect Alternative Splicing Events

Next-generation sequencing (NGS) transcriptomic technologies have led to an explosion of new ASE findings in the retina. Authors have detected almost 80,000 novel ASEs, identifying of around 30,000 novel exons, 28,000 exon skipping events and 22,000 novel alternative splice sites (Figure 1C) [35]. Approximately 25% of these events maintained the open reading frame (ORF), encoding novel protein-coding transcript isoforms which may be critical for the retinal function. Most recent examples of new ASE findings in the retina producing protein-coding transcripts include Dp71 isoforms, which have been described to be differentially expressed in the mouse brain and retina [36]. Isl1 alternative splicing also produces two different isoforms in the mouse retina [37] and an alternative splicing product of Otx2, which plays critical roles in retina development, has been detected in neural retinal and retinal pigmented epithelial cells [38]. 

However, new emergent technologies suggest that the number of ASEs generating transcript diversity is even higher than expected. Typical RNA-seq read lengths are <200 bp, and such short reads are often unable to resolve the full-length sequence of RNA transcripts and determine specific isoforms [39]. In contrast, long-read sequencing is exceptionally useful in comprehensive characterization of RNA isoforms [40]. 

This technology, as provided by PacBio (Menlo Park, CA, USA) and Oxford Nanopore (Oxford, UK), is able to detect and quantify isoforms by sequencing molecules end to end from 3′ polyA to 5′ cap, thereby allowing the full-length transcript identification. A single long read covering a full-length transcript can then accurately define its transcription start site, all splice sites and the polyA site [41].

Long-read sequencing has already been proved useful to detect a previously unannotated isoform of the retinal degeneration gene CRB1. Mutations in CRB1 can cause a spectrum of inherited retinal dystrophies (IRDs), including RP and Leber Congenital Amaurosis (LCA). Using long-read sequencing, it was revealed that the most abundant retinal CRB1 isoform, CRB1-B, was a previously unknown variant containing unconventional 5′ and 3′ exons [42]. In mice, Crb1-A and -B isoforms have different promoters that drive their expression in Müller glia and photoreceptors, respectively. CRB1 is required for integrity of outer limiting membrane (OLM) junctions between Müller glia and photoreceptors [43]. In contrast to deletion of Crb1-A, which does not alter retinal function [44], deletion of Crb1-B causes retinal degeneration, recapitulating the human phenotype [42]. 

### 2.3. Implications of Retinal Alternative Splicing in Functional Analyses of IRD Genes

The complexity of the retinal alternative splicing system has a great impact in the diagnosis of IRDs. Many sequencing studies can readily identify the first but fail to identify the second mutant allele, which may be located within an exon of an as yet unidentified transcript. Unannotated isoforms can encode uncharacterized protein-coding sequences or represent unknown expression patterns, overall causing disease-causative mutations to be either misinterpreted or completely missed out. 

Experiment design and interpretation to understand gene function is also impaired by lack of comprehensive isoform sequence information. Unless transcript sequences are known, it is difficult to be certain that a “knockout” mouse allele fully eliminates expression of all isoforms. This was the situation encountered when generating a mouse model for the retinal dystrophy Cerkl gene, whose mutations can cause RP and Cone-Rod Dystrophy. The CERKL locus exhibits an incredibly high transcriptional complexity in human and mouse with alternative first exons, alternatively spliced exons, intron retention and additional splice sites [45]. To approach the function of the gene, the authors decided to create a knockout mouse model by deleting the Cerkl first exon and proximal promoter [46]. However, unreported alternative promoters directed basal expression of Cerkl in the retina, resulting in a knockdown model. Subsequently, the authors generated another Cerkl model using CRISPR/Cas9 [47]. The main message being that the comprehension of ASEs in the retina is essential to increase the genetic diagnosis of IRDs as well as to analyze gene regulation and function.

### 2.4. Splicing Factors Involved in the Processing of Retinal Transcripts

The splicing process is carried out by a dynamic ribonucleoprotein (RNP) machinery, the spliceosome, which is characterized by the orchestrated assembly and disassembly of several small nuclear RNPs (snRNPs) and associated protein co-factors. Fundamentally, the spliceosomal core is comprised of five different snRNPs named U1, U2, U4, U5 and U6 after the small nuclear RNAs (snRNAs) that compose them [48]. 

The assembly of the spliceosome system is a stepwise process in which the formation of different complexes between snRNPs and other proteins in pre-mRNA occurs. First, U1 snRNP recognizes the 5′ splice donor site (SDS) of the intron through base pairing of the U1 snRNA with the pre-mRNA. In addition, U1C, a protein from U1 snRNP, stabilizes this interaction [49]. Next, intronic 3′ splice acceptor site (SAS) is identified by U2 snRNP, SF1 and U2AF (U2 snRNP associated factors) [50]. These steps lead to the formation of complex E (Figure 2A).

Subsequently, U2 snRNA recognizes pre-mRNA sequences around the adenosine at the BS by base pairing and engages there generating the complex A (Figure 1B) [51]. U2 snRNP interacts with U1 snRNP to define intron boundaries forming the intron definition complex [52]. Afterwards, U5 and U4/U6 snRNPs are joined together as U4/U6·U5 tri-snRNP and recruited to the spliceosome to conform the complex B (Figure 2C), which is catalytically inactive [53]. Then, complex B becomes active (complex B*) throughout a sequence of compositional and conformational rearrangements. As a consequence of the release of U4 and U1 snRNPs from the spliceosome, U6 snRNP replaces U1 snRNP in the SDS [54]. These rearrangements lead to the interaction between U2 and U6 snRNP which will catalyse the splicing reaction [55]. 

Thereupon, complex B* (the active form of complex B) undergoes the first catalytic step of splicing, constituting the complex C (Figure 2D). Then, a set of conformational rearrangements occur in complex C. After the second catalytic step of splicing, 5′ and 3′ exons are ligated, intronic pre-mRNA is released in form of RNP and spliceosomal snRNPs (U2, U5 and U6) dissociate to be recycled for next splicing reactions (Figure 2E) [56].

Among all ubiquitously expressed splicing factors, some are of special relevance for the processing of retinal transcripts since mutations in them cause retinal dystrophies (this will be discussed in detail in Section 3.1). Interestingly, most of these factors are essential for the interaction between U4/U6 and U5 snRNPs and the stabilization of the U4/U6·U5 tri-snRNP, such as PRPF3, PRPF4, PRPF6, PRPF8 and PRPF31. Other examples include SNRNP200 and DHX38, RNA helicases that mediate different rearrangements of the spliceosome. PAP1, whose specific function in splicing is not well determined, is also involved in retinal dystrophies (reviewed in [57]). Finally, mutations in the spliceosomal component *CWC27* (which interacts with human PRPF8 homolog in yeast [58]) cause a range of clinical phenotypes, including retinal degeneration [59]. Therefore, proper splicing of retinal transcripts is an important process for accurate retinal function and homeostasis, making the retina particularly sensitive to splicing alterations.

### 2.5. Regulation of the Splicing

AS can be further modulated through cis-regulatory elements and trans-acting splicing factors, which are tissue-specific and contribute to the generation of tissue-specific isoforms. Cis-acting regulatory elements, also known as splicing regulatory elements (SREs), are specific sequences in the pre-mRNA located near the splice site that can enhance or silence splicing. Trans-acting regulatory proteins are recruited by enhancer or silencer SREs in order to steady or destabilize spliceosome assembly, hence controlling the inclusion or omission of differentially spliced exons. Therefore, spliceosome assembly represents a key control point in deciding between constitutive and alternative splicing [60,61].

#### 2.5.1. Trans-Regulatory Elements: RNA-Binding Proteins

Processing of retinal transcripts is deeply regulated by means of a host of trans-acting specific regulatory proteins depending on the retinal cell type. These splicing regulatory proteins bind pre-mRNA to induce or supress different steps of the splicing machinery assembly and, concomitantly, activate or repress the inclusion of alternative exons. Depending on their function, we will distinguish between splicing activators and repressors [61]. 

In the retina, the splicing repressor PTBP1 is downregulated in photoreceptors and retinal neurons, whereas its homolog PTBP2, which regulates the inclusion of neuron-specific exons, is present in these cell types [62]. In fact, these splicing regulators act antagonistically, and downregulation of PTBP1 leads to the upregulation of PTBP2 [63]. 

Interestingly, some splicing regulators, such as RBFOX and NOVA1, are specific for some retinal neurons but not expressed in others, e.g., photoreceptors [62]. Typically, RBFOX protein family has been described to be related to synaptogenesis and neurogenesis in CNS neurons. Lately, RBFOX1, RBFOX2 and RBFOX3 have been also reported to contribute to splicing of retinal transcripts in amacrine, horizontal and ganglion cells. RBFOX1 and RBFOX2 may be important for visual function, particularly for depth perception, while RBFOX3 is not [64,65,66]. In contrast, photoreceptors present specific factors that differ from those typically found in neurons. A well-studied example is the Musashi protein family (MSI1 and MSI2). These splicing factors are expressed in neuronal tissues in which they control stem-cell renewal and supress cell differentiation. However, and specifically concerning the retina, MS1 and MS2 regulate the splicing of photoreceptor-differentially spliced exons and are essential for photoreceptor function and homeostasis [62,67]. In addition, the upregulation of MS1 synergically interacts with the downregulation of PTBP1 in order to process photoreceptor-specific transcripts [68]. Summarizing, splicing regulators are differentially expressed depending on the retinal cell type and interact with each other to regulate all cell-specific splicing programs.

#### 2.5.2. Cis-Regulatory Elements: Enhancers and Silencers

SREs, located in exons or introns, are able to promote (enhancers) or inhibit (silencers) splicing from neighboring splice sites [5]. There are at least four major types of cis-regulatory elements depending on their location and associated effect on splicing: exonic splicing enhancer (ESE), exonic splicing silencer (ESS), intronic splicing enhancer (ISE) and intronic splicing silencer (ISS). AS is often regulated by the combination of general and tissue-specific regulators. Furthermore, several disease-causing mutations that disrupt the cis-regulatory elements for splicing have been identified [69], indicating that they are critical for the function of the retina. 

ESEs promote the recognition of exons with weak splice sites by assisting in the recruitment of splicing factors to the adjacent intron [70,71]. As an example concerning a relevant retinal gene, an ESE located in the *Nr2e3* gene has been recently reported [21]. NR2E3 is a transcription factor necessary for retinal development and homeostasis. Mutations in this gene can cause either RP or Enhanced S-cone Syndrome. The *NR2E3* locus produces two different protein coding isoforms: the full-length isoform, containing the 8 exons of the gene, and a shorter isoform, which lacks exon 8 and the functional domains encoded in this exon. Both isoforms have been detected in the retina, but the proportion of each transcript may vary depending on the developmental stage. The predicted ESE, located in exon 8 of *Nr2e3*, most probably promotes the splicing between exons 7 and 8, thus facilitating the production of the full-length isoform. A deletion of the ESE causes an increase of intron 7 retention, producing an imbalance between the two isoforms that may be associated with retinal disease [21].

## 3. The Role of Alternative Splicing in Retinal Disease

Around 9% of all disease-causing mutations are estimated to alter pre-mRNA splicing [15,16]. These mutations can disrupt or alter cis-regulatory sequences, or the binding of trans-acting splicing factors. Mutations affecting cis-acting splice sites or regulatory sequences can lead to inappropriate exon skipping, intron inclusion, exon inclusion or activation of cryptic splice sites (some of them in deep intronic positions), usually leading to frameshifts and premature termination. Mutations causing IRDs can also affect unannotated isoforms, which may cause misinterpretation of the genetic diagnosis. Indeed, most of the genes causing IRDs undergo alternative splicing (Table 1 compiles alternative splicing events of IRD genes). Around 90% of the IRD genes produce more than one transcript (Figure 3A), and for most of them (54%), between two to 10 alternatively spliced transcripts have been reported. Indeed, the multiplicity of transcripts also impacts on the multiplicity of encoded proteins, and at least 85% of the IRD genes display several protein isoforms (Figure 3B). Most of the genes produce between two and five distinct coding transcripts, but 9% of them are able to generate more than 10 singular coding transcripts that translate into proteins that most probably carry out differential functions. Therefore, understanding AS and isoform sequence information is fundamental to comprehend both normal gene function and the phenotypic consequences of gene mutations.

### 3.1. Trans-Acting Mutations in Splicing Factors: PRPF31 

Although non-syndromic IRDs show a phenotype restricted to the eye, not all of the disease-causing genes (compiled in Table 1) are exclusively expressed in the retina. Mutations in the genes encoding splicing factors that are ubiquitously expressed and are important for the general process of pre-mRNA splicing–including PRPF3, PRPF4, PRPF6, PRPF8, PRPF31, SNRNP2000, PAP1, DHX38 and *CWC27* [57,59]—have been identified as causative of IRDs [69,72]. In fact, they represent the second most common cause for autosomal dominant RP (adRP), after mutations in the rhodopsin gene (RHO) [73]. It is unclear why mutations in these factors cause a phenotype restricted to the retina while being tolerated by other tissues. However, some of these factors are more highly expressed in the retina than in other tissues [74], which suggest higher splicing requirements for the retina. Moreover, photoreceptors exhibit a specific splicing program driven by the MS1 factor, which initiates during the development and affects transcripts encoding components of primary cilia and outer segments required for phototransduction [62]. Although the specific disease mechanisms are mostly unknown, some possible explanations have been proposed to account for the retina being highly sensitive to mutations that disturb the spliceosome assembly and function [75]. The reduced levels of splicing factors most probably lead to transcriptional dysregulation of specific retinal genes [57]. Furthermore, mutations in splicing factors induce protein-folding defects, which cause aggregation of misfolded mutant proteins [76]. Photoreceptor cells do not regenerate and thus, aggregates will accumulate over time resulting in increased probability of cell death. Alternatively, aggregates will activate the unfolded protein response (already detected in a RP model [77]), which will create long-lasting stress after constant detection of mis-folded proteins, ultimately triggering apoptosis [57]. 

PRPF31, an essential protein involved in the assembly and stability of the U4/U6·U5 tri-snRNP, illustrates the difficulty in studying splicing factors causing IRDs. Heterozygous mutations in PRPF31 gene have been determined as cause for adRP [78,79,80,81]. However, modifier loci can prevent disease development and thus, adRP due to PRPF31 mutations shows incomplete penetrance, resulting in individuals carrying PRPF31 mutations that do not present RP symptoms, even in the same family [82,83]. Although amino acid substitutions are the main mutation type found in splicing factor genes, most mutations in PRPF31 are deletions, frameshifts or mutations that alter splicing, leading to the introduction of premature stop codons and resulting in reduced PRPF31 levels [57]. In contrast, the PRPF31 Ala216Pro variant, presents a dominant negative effect, as the mutant protein shows a stronger interaction with PRPF6, which results in the inhibition of the protein-RNA movements and subsequent impairment of spliceosome activation and recycling of the proteins for future splicing events [84]. Interestingly, over-expression of PRPF6 rescues the mutant phenotype and, consequently, PRPF6 expression may represent an additional factor accounting for some cases of PRPF31 adRP incomplete penetrance [85]. Regarding the retinal phenotype, studies using induced pluripotent stem cell (iPSC)-derived organoids, revealed that impaired splicing is restricted to retinal cells only and it affects genes involved in RNA processing as well as genes involved in phototransduction and ciliogenesis, which have been associated with progressive degeneration and cellular stress [86,87]. In fact, mis-splicing of ciliary genes was associated with severe defects in the retinal pigmented epithelium (RPE) cells, which are typically affected (together with photoreceptors) in the RP disease [87].

### 3.2. Cis-Acting Mutations Altering the Splicing

Splice-site mutations have been identified in patients with RP, USH or Stargardt disease, among other IRDs. Mutations can either disrupt a consensus splice site sequence causing exon skipping, shift the splicing acceptor or donor splices sites, or promote the usage of cryptic deep intronic sequences that result in different exon size, intron retention, or novel exon inclusion. One of the current challenges in the genetic diagnosis of IRDs is the detection and functional validation of variants that have not been previously reported and whose functional significance remains unclear. New cis-acting mutations causing retina-specific splicing defects are usually tested in HEK293T cells using in vitro mini- and midi-genes splice assays because IRD genes are not commonly expressed in accessible human tissues [88,89,90]. However, HEK293T cells do not reproduce retinal cell conditions since they do not express retina-specific splicing factors and adjuvant proteins. For this reason, variants showing no effect on splicing in these assays may still be proven to be pathogenic when assessed in induced iPSC-derived photoreceptor precursors [91]. Cis-acting variants are commonly hypomorphic variants, which reduce the range of correct splicing and lead to splicing alterations while retaining considerable productive transcript [92,93]. The severity of these variants is evaluated according to the percentage of the wildtype (WT) remaining product (the higher the percentage of the WT allele, the lower the severity of the variant). Many of these hypomorphic variants are identified in ciliary genes that when bearing a more severe mutation cause syndromic ciliopathies.

#### 3.2.1. Non-Canonical Splice Site Variants (NCSS): ABCA4

Canonical splice sequences are located at the AG-receptor (−1 and −2) and GT-donor (+1 and +2) nucleotides, affecting directly the primary sequence of the receptor and donor sites, respectively. NCSS variants are instead located either at the first and last three nucleotides of an exon, or else at the −3 to −14 nucleotides from the acceptor site and +3 to +6 nucleotides from the donor site, altering the splicing motif recognition by the splicing factors. NCSS variants can lead to partial or entire exon skipping, producing partial in-frame deletions or open reading frame disruptions that cause frameshifts and lead to prematurely truncated proteins.

Stargardt disease is the most prevalent inherited macular dystrophy, usually presented as an autosomal recessive condition caused by mutations in the *ABCA4* gene. *ABCA4* is a large, highly polymorphic gene, consisting of 50 exons, which presents over 1200 disease-associated variants [94]. A recent study reported that 18% likely pathogenic variants present a significant splice site alteration, including NCSS and deep intronic variants [95,96]. The identification of splicing variants in a highly polymorphic gene such as *ABCA4* is not unusual, however, the prediction and confirmation of pathogenicity has proven difficult [97]. Several sequencing studies only find one pathogenic allele and fail to identify the second and, as a consequence, 15% of the cases remain “unsolved” [94]. Some authors propose that hypomorphic splice variants account for some of these *ABCA4* missing pathogenic alleles [92,93]. 

The NCSS variant c.161G>A, which has been previously associated with Stargardt disease [98], demonstrates the complexity of the *ABCA4* genetic analysis. The c.161G>A variant is located in the first nucleotide of exon 3, a coding region of *ABCA4*. Exon 3 shows a weak natural exon skipping in 14% of the WT transcripts [96]. Notably, the variant c.161G>A has two different effects: it causes exon 3 skipping in around 50% of the transcripts (p.Cys54Serfs*14) but it is also a missense mutation that alters the amino acid sequence of ABCA4 protein (p.Cys54Tyr) (Figure 4A). Therefore, both events are contributing to the pathogenicity of the variant. Surprisingly, this variant has also been observed in the “control” population database [96]. Common hypomorphic variants at the *ABCA4* locus alter risk properties and they can be pathogenic only when in trans with a loss-of-function *ABCA4* allele [93,99]. Therefore, occasionally, they result in disease expression, particularly in those patients who carry only a pathogenic allele, thus explaining why some variants are found to be pathogenic in some individuals but not in others. Another example is the NCSS variant c.4849-8C>G, which has also been proved to be pathogenic since it lowers the value of the Pyrimidine tract upstream of the 3′ splice site of intron 34, thus producing transcripts with intron retention that leads to premature protein truncation (Figure 4B) [88].

#### 3.2.2. Deep Intronic Variants: ABCA4, CEP290 and USH2A

Deep intronic variants are located more than 100 bp away from exon-intron junctions, which usually lead to pseudo-exon inclusion due to activation of novel splice sites. The introduction of a pseudo-exon (PE) commonly alters the reading frame introducing a premature stop codon, which targets the mutant mRNA for degradation by NMD [100].

LCA is an IRD that results in severe visual loss in early childhood. One of the most common causative LCA genes is *CEP290*, encoding a centrosomal protein, which has also been associated with syndromic ciliopathies [101]. The most common *CEP290* mutation is the deep intronic c.2991+1655A>G variant, which is found in the majority of the *CEP290* LCA patients (86%) [102]. The mutation creates a strong splice donor site (SDS) that induces the inclusion of a cryptic exon between exons 26 and 27. This cryptic exon encodes a premature stop codon (p.Cys998Stop) (Figure 4C) [103]. Remarkably, 50% of the product is still spliced correctly, which may be sufficient for its function in other organs but not in photoreceptors [102,103], thereby highlighting the importance of splicing in the retina and explaining the retina-only phenotype of this particular mutation.

*USH2A* is the most commonly mutated gene in USH type 2, characterized by congenital hearing impairment and RP. The deep intronic variant c.7595-2144A>G is the second most common cause of USH type 2A [104,105]. This variant creates a novel SD site in intron 40, leading to the insertion of a PE into the mature transcript. This PE encodes a premature termination of translation (p.Lys2532Thrfs*56) (Figure 4D) [104,105]. In contrast to the case of the deep intronic variant c.1938-619A>G of the *ABCA4* gene, which created a novel splice site, this mutation in *USH2A* strengthens a cryptic splicing site, probably by increasing the strength of ESE motifs that induce the inclusion of the PE (Figure 4E) [96].

#### 3.2.3. Deep Exonic Variants: RHO

Interestingly, mutations located in the middle of an exon can also affect splicing, as it occurs with the c.620T>G variant in rhodopsin, a light-sensitive receptor involved in rod visual phototransduction. Mutations in *RHO* are the most common cause for adRP [106]. Among them, the c.620T>G variant, located in exon 3, was first classified as a missense mutation (Met207Arg) causing severe early-onset adRP [107,108]. The number of altered amino acids in mutations affecting Met207 and surrounding residues usually correlates with the severity of the adRP phenotype. However, another variant affecting the same nucleotide, the c.620T>A (Met207Lys), was associated with a mild late-onset adRP [109]. Researchers have recently discovered that the initial c.620T>G variant (previously classified as missense) is in fact a splicing mutation which generates a particularly strong splice acceptor that results in a 90 bp in-frame deletion and subsequent mislocalization of rhodopsin in photoreceptors [110], thus explaining the severe phenotype found in individuals carrying this mutation (Figure 4F). This finding suggest that point mutations located in exons should be routinely evaluated in silico and subsequently tested for their potential disruptive effect in mRNA splicing in order to avoid misinterpretation of the variants and understand genotype-phenotype correlations, disease mechanisms and ultimately predict the disease course.

### 3.3. Mutations in Retina-Specific Exons and Microexons: BBS8, RPGR and DYNC2H1

As discussed before, some mutations in widely expressed genes (e.g. in splicing factor genes or CEP290) result in primarily ocular disease. That is also the case of mutations that affect the prevalence of retina-specific transcripts or mutations in retina-specific exons [69]. Identifying retina-specific transcripts is thus essential to increase the genetic diagnosis yield in IRDs as well as to design specific therapeutic approaches.

Mutations in the RPGR gene, which encodes a ciliary protein, have been identified as the cause of over 70% of X-linked RP (XLRP). RPGR undergoes extensive splicing (Table 1) and several transcripts for this gene have been identified [111,112,113], among them the constitutive transcript, that contains exons 1 to 19, and a retina-specific transcript, which contains constitutive 1–14 exons plus an alternative 3′ terminal exon known as ORF15 [113]. All documented RPGR mutations responsible for XLRP affect the RPGR^ORF15^ transcript, and 80% of these mutations occur in exon ORF15, which has been identified as a mutational hotspot [113]. The expression of both the constitutive and the retina-specific isoforms is regulated during retinal development and, interestingly, overexpression of the constitutive isoform causes retinal degeneration in mouse, suggesting that the balance between both isoforms is necessary for correct retinal function [114]. 

Some genes causing syndromic diseases can also contribute to the development of a retinal disease. Such is the case of BBS8, mutated in several ciliopathies, which presents an alternative 30 bp microexon, exon 2a, that results in a 10 amino acid longer protein whose expression is exclusively restricted to photoreceptors. The inclusion of this microexon is due to specific ISEs exclusively recognized by splicing factors of photoreceptor cells [115,116]. Surprisingly, the A>G substitution (IVS1-2AG), located in the canonical 3′ AG-acceptor of exon 2a, forces the use of a cryptic splice site located 7 nt downstream of the mutated site, which probably results in premature termination of the BBS8 reading frame and elimination of the protein in photoreceptors [116]. Cell types other than photoreceptors do not recognize exon 2a and are not affected therefore by the IVS1-2AG mutation, explaining the RP-restricted phenotype.

A similar case occurs in the DYNC2H1 gene and has been recently associated with severe ciliopathies. DYNC2H1 contains a microexon of 21 bp that is predominant in retinal transcripts [35,117]. Authors hypothesize that the isoform containing the microexon could be the major isoform expressed in photoreceptors because its expression in retinal organoids increase as photoreceptors differentiate, becoming the dominant transcript when photoreceptors are mature [117]. The DYNC2H1 c.9836C>G mutation is predicted to introduce a premature stop codon in the microexon, possibly resulting in a severely truncated protein. As in previous cases, this variant causes nonsyndromic retinal degeneration, which suggest that the canonical isoform, expressed in all the other tissues, remains unaffected [117]. All these cases strongly indicate that AS is the main mechanism through which mutated syndromic ciliopathy genes lead to non-syndromic IRDs and highlight the importance of identifying retina-specific transcripts that are undeniably important for visual function.

## 4. Therapeutic Strategies to Modulate Aberrant Splicing

The eye is an ideal target organ for therapeutic interventions due to its easy accessibility and the presence of a blood-retina barrier that prevents exchange of the therapeutic molecules with other organs, thus reducing side effects and undesirable immune responses. Splicing modulation has been a key target for new therapeutic strategies to treat IRDs. 

As aforementioned, U1 splice factor binds complementarily with nucleotides at the exon-intron border, promoting the recognition of splice donor site (SDS) and initiation of the splice process [118]. Aberrant splicing in IRDs is often the result of disturbed U1 binding to mutated SDS. Therefore, mutation-adapted U1 can be designed to match all nucleotide of patient SDS (including the mutation), leading to correction of splice defects. This strategy has been proven useful for RHO [119] and RPGR mutations [120]. The main advantage of the U1 technique is that it corrects the endogenously expressed transcript and reduces the amount of mutated protein, which is especially important for the treatment of dominant diseases with gain of function mutations [120].

Spliceosomal-mediated RNA trans-splicing (SMaRT) has also been considered for therapeutic approaches [121,122]. Unlike cis-splicing, trans-splicing naturally joins exons from two independent pre-mRNA molecules and results in a final mRNA consisting of the 5′ part of the first pre-mRNA and the 3′ part of the second pre-mRNA [123]. SMaRT requires the introduction of an exogenous pre-mRNA trans-splicing molecule (PTM), which consist of a binding domain to target the endogenous mutated pre-mRNA, an artificial intron containing the elements necessary for splicing and the cDNA gene sequence to be repaired. SMaRT technology producing hybrid mRNAs has been used as a therapeutic tool for correcting RHO [124] and CEP290 [125] mutations. As expected, the replacement of the mutated sequence decreases the mutant protein synthesis, which is important in cases of dominant IRDs (RHO) and increases the level of correct protein levels in recessive mutations (CEP290). 

RNA therapeutic strategies for treating IRDs have been recently reviewed [126]. In this context, the use of siRNA and shRNA agents is worth nothing. siRNAs have shown potential in patients with age-related macular dystrophy (AMD) [127], however, noninternalized siRNAs may stimulate the immune system via Toll-like receptor activation in the RPE, thus inducing retinal degeneration [128,129]. On the other hand, shRNAs have been proven particularly beneficial in the treatment of autosomal dominant disorders, such as those caused by RHO mutations [130] as well as in silencing VEGF production in AMD mouse models [131]. One of the most promising therapeutic agents are antisense oligonucleotides (AONs), small RNA molecules that bind complementarily to the pre-mRNA to correct aberrant splicing caused by the activation of cryptic splice sites [132,133,134]. AON-based therapies have shown promising results for mutations in CEP290 [135,136,137], OPA1 [138], CHM [139], USH2A [105] and ABCA4 [91,140], and are now being tested in clinical trials with patients.

Finally, gene editing techniques allow direct correction of the pathogenic allele in the genomic DNA. CRISPR-Cas9 has been successfully used to correct the splicing effect of the deep intronic CEP290 c.2991 + 1655A > G mutation in vivo, first in mouse and now already in human clinical trials [137]. This approach could be useful to correct aberrant splicing caused by deep intronic mutations in other IRD genes. 

## 5. Conclusions

The retina, as an extension of the nervous system, exhibits an exceptional transcript diversity that require an exquisite regulation performed by general and specific splicing factors. Some retinal cells, such as post-synaptic neurons and photoreceptors, express different splicing factors and, interestingly, mutations in splicing factors ubiquitously expressed and important for the general process of pre-mRNA splicing can cause a retina-restricted phenotype.

Among all the alternative splicing events identified in the retina, incorporation of cassette exons and microexons strongly contributes to development and homeostasis. The retina is one of the tissues that present a higher number of ASEs. In fact, identification of these events has been increasingly growing as massive parallel sequencing and other emergent technologies are developed and implemented in routine genetic diagnosis.

Around 90% of IRD genes present more than one transcript and a considerable percentage of mutations alter splicing or are located in unidentified transcripts exclusively expressed in the retina. Therefore, lack of information regarding AS may cause misinterpretation of IRD diagnosis. One of the current challenges is the detection of variants that have not been previously reported, including hypomorphic variants that are also found in the general population and whose functional significance remains unclear. Functional analysis of mutations causing retina-specific splicing defects should be tested in retinal cell-like environments, such as organoids or iPSC-derived photoreceptor precursors, since some of the splicing factors required are only expressed in the retina. In specialized organs and tissues such as the retina, comprehensive isoform information is fundamental to increase genetic diagnosis yield and comprehend both normal gene function and phenotypic consequences of mutations.

Modulation of alternative splicing in the retina is also crucial to develop novel therapeutic approaches. In fact, siRNAs and AONs are mutation-specific therapies that represent a promising tool to treat retinal dystrophies in a patient-focused personalized medicine.

## Figures and Tables

**Figure 1 ijms-22-01855-f001:**
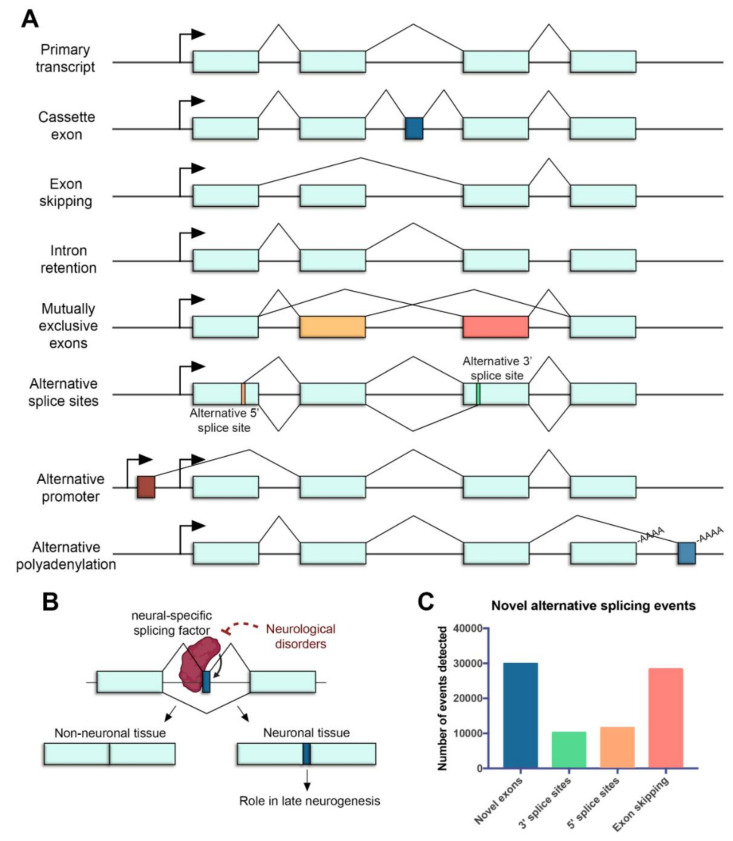
Alternative splicing in the neural retina. (**A**) Common mechanisms of alternative splicing in the retina. Boxes represent exons, lines represent introns, promoters are represented with arrows and polyadenylation sites are indicated with -AAAA. Exon regions included in the alternative transcript are colored. (**B**) Microexons have a role in late neurogenesis and are relevant in neurological disorders. The reduced expression of neural-specific splicing factors that regulate the inclusion of microexons is linked to the altered splicing of microexons in patients with neurological disorders. (**C**) Novel alternative splicing events in the human retina detected by RNA sequencing (data from [35]).

**Figure 2 ijms-22-01855-f002:**
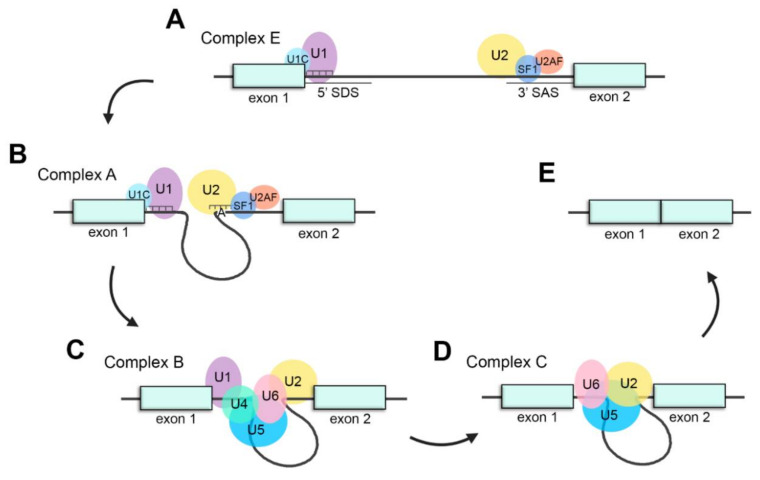
Schematic representation of the splicing process. (**A**) Assembly of the spliceosome: U1 snRNP recognizes the splice donor site (SDS) and U2 snRNP recognizes the splice acceptor site (SAS) to generate complex E. (**B**) U2 recognizes the adenosine at the branch-site and forms complex A. (**C**) The U4/U6·U5 tri-snRNP joins the spliceosome to form complex B. (**D**) U4 and U1 are released, U6 replaces U1 recognizing the SDS and interacts with U2, generating complex C and catalyzing the splicing reaction. (**E**) Exons are ligated, and intronic pre-mRNA and spliceosomal snRNPs are liberated.

**Figure 3 ijms-22-01855-f003:**
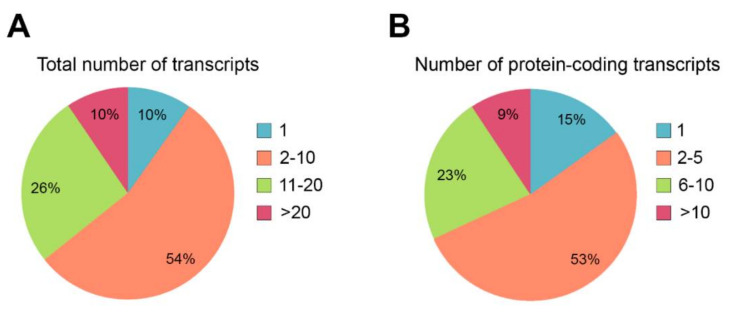
Quantification of alternative splicing events described in genes causing IRDs. (**A**) Percentage of IRD genes presenting alternative splicing events. Only 10% of the IRD genes produces a unique transcript, and most genes (54%) generate between 2 and 10 transcripts. (**B**) Percentage of IRD genes showing diverse protein-coding transcripts. Only 15% of the IRD genes produce one protein isoform. Most of the genes (53%) produce between 2 and 5 diverse protein isoforms. The list of genes was obtained from https://sph.uth.edu/retnet/ (accessed on 20 January 2021). Information about the number and type of the transcripts was obtained from https://www.ensembl.org/ (accessed on 20 January 2021).

**Figure 4 ijms-22-01855-f004:**
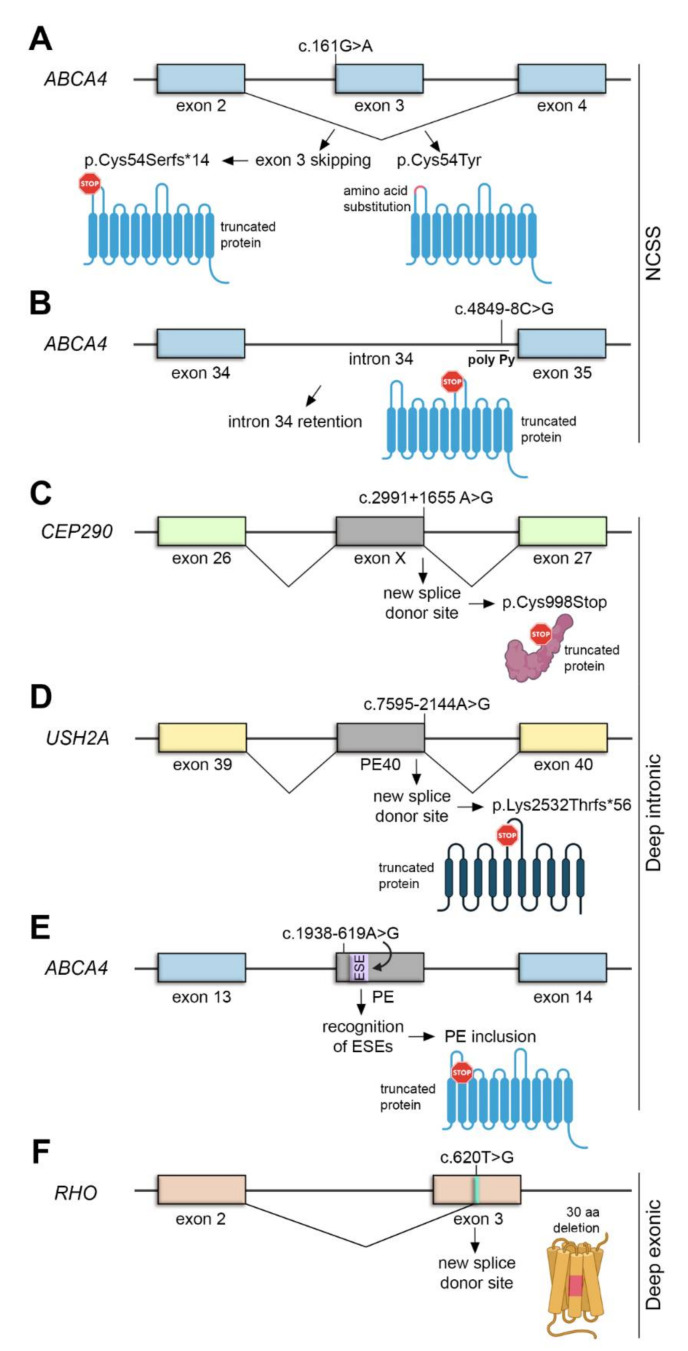
Overview of cis acting mutations altering splicing: NCSS (**A**,**B**), deep intronic (**C**–**E**) and deep exonic variants (**F**). (**A**) *ABCA4* exon 3 shows a weak natural exon skipping. The c.161G>A mutation increase exon 3 skipping (producing a truncated ABCA4 protein) and the p.Cys54Tyr amino acid substitution. (**B**) The *ABCA4* c.4849-8C>G mutation lowers the value of the poly-Py tract, thus causing intron 34 retention and production of a truncated protein. (**C**) The *CEP290* c.2991+1655A>G mutation creates a new SDS, that induce inclusion of a cryptic exon (exon X) encoding a premature stop codon. (**D**) The *USH2A* c.7595-2144A>G mutation creates a new SDS that causes pseudoexon inclusion (PE40) and introduces a premature stop codon. (**E**) The *ABCA4* c.1938-619A>G mutation, located in a cryptic pseudoexon (PE), leads to the recognition of ESEs that promote PE inclusion, leading to the truncation of the protein. (**F**) The *RHO* c.620T>G mutation creates a strong splice donor site that results in a 30 amino acid in-frame deletion.

**Table 1 ijms-22-01855-t001:** List of genes causative of IRDs (https://sph.uth.edu/retnet/ (accessed on 20 January 2021)) showing location, associated disease, number of splice variants and coding transcripts. Genes mentioned in the main text are indicated in bold.

Gene	Location	Associated Disease	Splice Variants	Coding Transcripts
***ABCA4***	**1p22.1**	**Recessive Stargardt disease, juvenile and late onset; recessive macular dystrophy; recessive retinitis pigmentosa; recessive fundus flavimaculatus; recessive cone-rod dystrophy**	**8**	**3**
*ABCC6*	16p13.11	Recessive pseudoxanthoma elasticum; dominant pseudoxanthoma elasticum	9	4
*ABHD12*	20p11.21	Recessive syndromic PHARC; recessive Usher syndrome, type 3-like	24	17
*ACBD5*	10p12.1	Recessive cone-rod dystrophy with psychomotor delay	8	7
*ACO2*	22q13.2	Recessive optic atrophy; recessive cerebellar degeneration with optic atrophy	7	2
*ADAM9*	8p11.23	Recessive cone-rod dystrophy	9	3
*ADAMTS18*	16q23.1	Recessive Knobloch syndrome; recessive retinal dystrophy, early onset	9	3
*ADGRA3*	4p15.2	Recessive retinitis pigmentosa	13	4
*ADGRV1*	5q14.3	Recessive Usher syndrome, type 2; dominant/recessive febrile convulsions	37	10
*ADIPOR1*	1q32.1	Recessive retinitis pigmentosa, syndromic, Bardet-Biedl like; dominant retinitis pigmentosa	5	4
*AFG3L2*	18p11.21	Dominant optic atrophy, non-syndromic; dominant spinocerebellar ataxia; recessive spastic ataxia	4	2
*AGBL5*	2p23.3	Recessive retinitis pigmentosa	10	7
*AHI1*	6q23.3	Recessive Joubert syndrome	17	10
*AHR*	7p21.1	Recessive retinitis pigmentosa	7	2
*AIPL1*	17p13.2	Recessive Leber congenital amaurosis; dominant cone-rod dystrophy	11	10
*ALMS1*	2p13.1	Recessive Alström syndrome	12	4
*ARHGEF18*	19p13.2	Recessive retinitis pigmentosa	6	5
*ARL2BP*	16q13.3	Recessive retinitis pigmentosa	4	3
*ARL3*	10q24.32	Dominant retinitis pigmentosa	1	1
*ARL6*	3q11.2	Recessive Bardet-Biedl syndrome; recessive retinitis pigmentosa	9	6
*ARMS2*	10q26.13	Age-related macular degeneration, complex etiology	1	1
*ARSG*	17q24.2	Recessive Usher syndrome, atypical	9	4
*ASRGL1*	11q12.3	Recessive retinal degeneration	11	6
*ATF6*	1q23.3	Recessive achromatopsia	2	1
*ATOH7*	10q21	Recessive nonsyndromal congenital retinal nonattachment	1	1
*ATXN7*	3p14.1	Dominant spinocerebellar ataxia w/ macular dystrophy or retinal degeneration	20	7
*BBIP1*	10q25.2	Recessive Bardet-Biedl syndrome	15	10
*BBS1*	11q13	Recessive Bardet-Biedl syndrome; recessive retinitis pigmentosa	25	7
*BBS10*	12q21.2	Recessive Bardet-Biedl syndrome	1	1
*BBS12*	4q27	Recessive Bardet-Biedl syndrome	3	3
*BBS2*	16q13	Recessive Bardet-Biedl syndrome; recessive retinitis pigmentosa	21	5
*BBS4*	15q24.1	Recessive Bardet-Biedl syndrome	18	5
*BBS5*	2q31.1	Recessive Bardet-Biedl syndrome	6	2
*BBS7*	4q27	Recessive Bardet Biedl syndrome	6	3
***BBS8***	**14q31.3**	**Recessive Bardet Biedl syndrome**	**13**	**7**
*BBS9*	7p14.3	Recessive Bardet Biedl syndrome	29	14
*BEST1*	11q12.3	Dominant macular dystrophy, Best type; dominant vitreoretinochoroidopathy; recessive bestrophinopathy; recessive retinitis pigmentosa; dominant retinitis pigmentosa	8	4
*C12orf65*	12q24.31	Recessive spastic paraplegia, neuropathy and optic atrophy	8	6
*C1QTNF5*	11q23.3	Dominant macular dystrophy, late onset; dominant macular dystrophy with lens zonules	4	3
*C2*	6p21.32	Age-related macular degeneration, complex etiology	17	11
*C3*	19p13.3	Age-related macular degeneration, complex etiology	18	5
*C8orf37*	8q22.1	Recessive cone-rod dystrophy; recessive retinitis pigmentosa with early macular involvement; recessive Bardet-Biedl syndrome	1	1
*CA4*	17q23.2	Dominant retinitis pigmentosa	6	4
*CABP4*	11q13.1	Recessive congenital stationary night blindness; recessive congenital cone-rod synaptic disease; recessive Leber congenital amaurosis	7	2
*CACNA1F*	Xp11.23	X-linked congenital stationary night blindness, incomplete; AIED-like disease; severe congenital stationary night blindness; X-linked progressive cone-rod dystrophy	6	4
*CACNA2D4*	12p13.33	Recessive cone dystrophy	26	9
*CAPN5*	11q13.5	Dominant neovascular inflammatory vitreoretinopathy	8	5
*CC2D2A*	4p15.33	Recessive retinitis pigmentosa and mental retardation; recessive Joubert syndrome	23	13
*CCT2*	12q15	Recessive Leber congenital amaurosis	14	3
*CDH23*	10q22.1	Recessive Usher syndrome, type 1d; recessive deafness without retinitis pigmentosa; digenic Usher syndrome with PCDH15	19	14
*CDH3*	16q22.1	Recessive macular dystrophy, juvenile with hypotrichosis	10	4
*CDHR1*	10q23.1	Recessive cone-rod dystrophy	7	4
*CEP164*	11q23.3	Recessive nephronophthisis with retinal degeneration	14	7
*CEP19*	3q29	Recessive Bardet-Biedl syndrome	2	2
*CEP250*	20q11.22	Recessive Usher syndrome, atypical	14	8
***CEP290***	**12q21.32**	**Recessive Senior-Loken syndrome; recessive Joubert syndrome; recessive Leber congenital amaurosis; recessive Meckel syndrome**	**34**	**13**
*CEP78*	9q21.2	Recessive cone-rod dystrophy with hearing loss; recessive Usher syndrome, atypical	24	14
***CERKL***	**2q31.3**	**Recessive retinitis pigmentosa; recessive cone-rod dystrophy with inner retinopathy**	**14**	**5**
*CFAP410*	21q22.3	Recessive cone-rod dystrophy	7	3
*CFB*	6p21.32	Age-related macular degeneration, complex etiology	14	4
*CFH*	1q31.3	Age-related macular degeneration, complex etiology; recessive drusen, early-onset	6	3
***CHM***	**Xq21.2**	**Choroideremia**	**5**	**2**
*CIB2*	15q25.1	Recessive Usher syndrome, type 1J	10	7
*CISD2*	4q22-q24	Recessive Wolfram syndrome	5	2
*CLCC1*	1p13.3	Recessive retinitis pigmentosa, severe	30	21
*CLN3*	16p11.2	Recessive Batten disease (ceroid-lipofuscinosis, neuronal 3), juvenile	62	20
*CLRN1*	3q25.1	Recessive Usher syndrome, type 3; recessive retinitis pigmentosa	8	4
*CLUAP1*	16p13.3	Recessive Leber congenital amaurosis	12	7
*CNGA1*	4p12	Recessive retinitis pigmentosa	7	6
*CNGA3*	2q11.2	Recessive achromatopsia; recessive cone-rod dystrophy; protein: cone photoreceptor cgmp-gated cation channel alpha subunit [Gene]	4	2
*CNGB1*	16q21	Recessive retinitis pigmentosa	9	6
*CNGB3*	8q21.3	Recessive achromatopsia Pingelapese; recessive progressive cone dystrophy	3	2
*CNNM4*	2q11.2	Recessive cone-rod dystrophy and amelogenesis imperfecta syndrome	4	1
*COL11A1*	1p21.1	Dominant Stickler syndrome, type II; dominant Marshall syndrome	14	10
*COL2A1*	12q13.11	Dominant Stickler syndrome, type I; dominant bone dysplasias, developmental disorders, osteoarthritic diseases, and syndromic disorders	9	2
*COL9A1*	6q13	Recessive Stickler syndrome; dominant multiple epiphyseal dysplasia (MED)	11	3
***CRB1***	**1q31.3**	**Recessive retinitis pigmentosa with para-arteriolar preservation of the RPE (PPRPE); recessive retinitis pigmentosa; recessive Leber congenital amaurosis; dominant pigmented paravenous chorioretinal atrophy**	**11**	**7**
*CRX*	19q13.32	Dominant cone-rod dystrophy; recessive, dominant and de novo Leber congenital amaurosis; dominant retinitis pigmentosa	7	4
*CSPP1*	8q13.1-q13.2	Recessive Jobert syndrome	16	8
*CTNNA1*	5q31.2	Dominant macular dystrophy, butterfly-shaped	44	27
***CWC27***	**5q12.3**	**Retinitis pigmentosa with or without skeletal anomalies**	**5**	**2**
*CYP4V2*	4q35.2	Recessive Bietti crystalline corneoretinal dystrophy; recessive retinitis pigmentosa	4	1
*DHDDS*	1p36.11	Recessive retinitis pigmentosa	25	16
***DHX38***	**16q22.2**	**Recessive retinitis pigmentosa, early onset with macular coloboma**	**14**	**6**
*DMD*	Xp21.2-p21.1	Oregon eye disease (probably)	32	20
*DNM1L*	22q12.1-q13.1	Dominant optic atrophy	30	11
*DRAM2*	1p13.3	Recessive macular dystrophy, early adult onset	11	2
*DTHD1*	4p14	Recessive Leber congenital amaurosis with myopathy	6	4
***DYNC2H1***	**11q22.3**	**Syndromic and non syndromic retinal degeneration**	**11**	**5**
*EFEMP1*	2p16.1	Dominant radial, macular drusen; dominant Doyne honeycomb retinal degeneration (Malattia Leventinese)	14	10
*ELOVL1*	1p34.2	Dominant optic atrophy, deafness, ichthyosis and neuronal disorders	16	3
*ELOVL4*	6q14.1	Dominant macular dystrophy, Stargardt-like; recessive spinocerebellar ataxia; recessive ichthyosis, quadriplegia and retardation	1	1
*EMC1*	1p36.13	Recessive retinitis pigmentosa	13	5
*ERCC6*	10q11.23	Age-related macular degeneration, complex etiology; Cockayne syndrome, recessive	12	4
*ESPN*	1p36.31	Recessive Usher syndrome	17	14
*EXOSC2*	9q34.12	Recessive retinitis pigmentosa with hearing loss and additional disabilities	11	6
*EYS*	6q12	Recessive retinitis pigmentosa	11	5
*FAM161A*	2p15	Recessive retinitis pigmentosa	7	2
*FBLN5*	14q32.12	Familial macular dystrophy, age-related	9	4
*FLVCR1*	1q32.3	Recessive retinitis pigmentosa with posterior column ataxia (PCARP)	5	2
*FSCN2*	17q25.3	Dominant retinitis pigmentosa; dominant macular dystrophy	3	2
*FZD4*	11p13-p12	Dominant familial exudative vitreoretinopathy	1	1
*FZD4*	11q14.2	Dominant familial exudative vitreoretinopathy	1	1
*GDF6*	8q22.1	Recessive Leber congenital amaurosis; dominant Klippel-Feil syndrome; dominant microphthalmia	3	3
*GNAT1*	3p21.31	Dominant congenital stationary night blindness, Nougaret type; recessive congenital stationary night blindness	5	3
*GNAT2*	1p13.3	Recessive achromatopsia	2	2
*GNB3*	12p13.31	Recessive congenital stationary night blindness	10	6
*GNPTG*	16p13.3	Recessive retinitis pigmentosa and skeletal abnormalities; recessive mucolipidosis III gamma	9	3
*GPR179*	17q12	Recessive complete congenital stationary night blindness	1	1
*GRK1*	13q34	Recessive congenital stationary night blindness, Oguchi type	3	1
*GRM6*	5q35.3	Recessive congenital stationary night blindness	6	3
*GUCA1A*	6p21.1	Dominant cone dystrophy; dominant cone-rod dystrophy	2	2
*GUCA1B*	6p21.1	Dominant retinitis pigmentosa; dominant macular dystrophy	1	1
*GUCY2D*	17p13	Dominant central areolar choroidal dystrophy	2	1
*GUCY2D*	17p13.1	Recessive Leber congenital amaurosis; dominant cone-rod dystrophy	2	1
*HARS1*	5q31.3	Recessive Usher syndrome	30	13
*HGSNAT*	8p11.21-p11.1	Recessive retinitis pigmentosa, non-syndromic; recessive mucopolysaccharidosis	10	4
*HK1*	10q22.1	Dominant retinitis pigmentosa; recessive nonspherocytic hemolytic anemia; recessive hereditary neuropathy (Russe type)	18	10
*HMCN1*	1q25.3-q31.1	Dominant macular dystrophy, age-related	5	2
*HMX1*	4p16.1	Recessive oculoauricular syndrome	2	2
*HTRA1*	10q26.13	Age-related macular degeneration, complex etiology	3	3
*IDH3B*	20p13	Recessive retinitis pigmentosa	12	4
*IFT140*	16p13.3	Recessive Mainzer-Saldino syndrome; recessive retinitis pigmentosa; recessive Leber congenital amaurosis	11	5
*IFT172*	2p33.3	Recessive Bardet-Biedl syndrome; recessive retinitis pigmentosa	34	6
*IFT27*	22q12.3	Recessive Bardet-Biedl syndrome	12	5
*IFT81*	12q24.11	Recessive cone-rod dystrophy; recessive spectrum of ciliopathies including retinal dystrophy	9	4
*IMPDH1*	7q32.1	Dominant retinitis pigmentosa; dominant Leber congenital amaurosis	18	11
*IMPG1*	6q14.1	Dominant macular dystrophy, vitelliform; recessive macular dystrophy, vitelliform; dominant retinitis pigmentosa	4	4
*IMPG2*	3q12.3	Recessive retinitis pigmentosa	1	1
*INPP5E*	9q34.3	Recessive Joubert syndrome; recessive MORM syndrome	6	3
*INVS*	9q31.1	Recessive Senior-Loken syndrome; recessive nephronophthisis	7	3
*IQCB1*	3q13.33	Recessive Senior-Loken syndrome; recessive Leber congenital amaurosis	7	5
*ITM2B*	13q14.2	Dominant retinal dystrophy; dominant dementia, familial	11	4
*JAG1*	20p12.2	Dominant Alagille syndrome	9	2
*KCNJ13*	2q37.1	Dominant vitreoretinal degeneration, snowflake; recessive Leber congenital amaurosis	5	5
*KCNV2*	9p24.2	Recessive cone dystrophy with supernormal rod electroretinogram	1	1
*KIAA1549*	7q34	Recessive retinitis pigmentosa; protein: KIAA1549 protein	2	2
*KIF11*	10q23.33	Dominant microcephaly, lymphedema and chorioretinopathy	1	1
*KIZ*	20p11.23	Recessive retinitis pigmentosa	15	8
*KLHL7*	7p15.3	Dominant retinitis pigmentosa	13	5
*LAMA1*	18p11.31-p11.23	Recessive retinal dystrophy and cerebellar dysplasia	9	2
*LCA5*	6q14.1	Recessive Leber congenital amaurosis	3	3
*LRAT*	4q32.1	Recessive retinitis pigmentosa, severe early-onset; recessive Leber congenital amaurosis	8	3
*LRIT3*	4q25	Recessive congenital stationary night blindness	2	2
*LRP5*	11q13.2	Dominant familial exudative vitreoretinopathy; dominant high bone mass trait; recessive osteoporosis-pseudoglioma syndrome; recessive familial exudative vitreoretinopathy	7	2
*LZTFL1*	3p21.31	Recessive Bardet-Biedl syndrome with developmental anomalies	16	5
*MAK*	6p24.2	Recessive retinits pigmentosa	7	5
*MAPKAPK3*	3p21.2	Dominant Martinique retinal dystrophy and retinitis pigmentosa	8	6
*MERTK*	2q13	Recessive retinitis pigmentosa; recessive rod-cone dystrophy, early onset	7	5
*MFN2*	1p36.22	Dominant optic atrophy with neuropathy and myopathy; dominant Charcot-Marie-Tooth disease	33	17
*MFRP*	11q23.3	Recessive microphthalmos and retinal disease syndrome; recessive nanophthalmos	6	3
*MFSD8*	4q28.2	Recessive macular dystrophy	62	24
*MKKS*	20p12.2	Recessive Bardet-Biedl syndrome	4	3
*MKS1*	17q22	Recessive Bardet-Biedl syndrome; recessive Meckel syndrome	13	7
*MMP19*	12q13.13-q14.3	Dominant cavitary optic disc anomalies	9	3
*MT-ATP6*	mitochondrion	Retinitis pigmentosa with developmental and neurological abnormalities; Leigh syndrome; Leber hereditary optic neuropathy	1	1
*MT-TH*	mitochondrion	Pigmentary retinopathy and sensorineural hearing loss	1	-
*MT-TL1*	mitochondrion	Macular pattern dystrophy with type II diabetes and deafness	1	-
*MT-TP*	mitochondrion	Retinitis pigmentosa with deafness and neurological abnormalities	1	-
*MT-TS2*	mitochondrion	Retinitis pigmentosa with progressive sensorineural hearing loss	1	-
*MTTP*	4q23	Recessive abetalipoproteinemia	11	5
*MVK*	12q24.11	Recessive retinitis pigmentosa; recessive mevalonic aciduria; recessive hyper-igd syndrome	17	10
*MYO7A*	11q13.5	Recessive Usher syndrome, type 1b; recessive congenital deafness without retinitis pigmentosa; recessive atypical Usher syndrome (USH3-like)	14	8
*NBAS*	2p24.3	Recessive optic atrophy and retinal dystrophy, syndromic;	9	7
*NDP*	Xp11.3	Norrie disease; familial exudative vitreoretinopathy; Coats disease	3	2
*NEK2*	1q32.3	Recessive retinitis pigmentosa; protein: NIMA (never in mitosis gene A)-related kinase 2 [Gene]	5	3
*NEUROD1*	2q31.3	Recessive retinitis pigmentosa	2	1
*NMNAT1*	1p36.22	Recessive Leber congenital amaurosis	5	3
*NPHP1*	2q13	Recessive Senior-Loken syndrome; recessive nephronophthisis, juvenile; recessive Joubert syndrome; recessive Bardet-Biedl syndrome	22	12
*NPHP3*	3q22.1	Recessive Senior-Loken syndrome; recessive nephronophthisis, adolescent	11	3
*NPHP4*	1p36.31	Recessive Senior-Loken syndrome, recessive nephronophthisis	11	2
***NR2E3***	**15q23**	**Recessive enhanced S-cone syndrome (ESCS); recessive retinitis pigmentosa in Portuguese Crypto Jews; recessive Goldmann-Favre syndrome; dominant retinitis pigmentosa; combined dominant and recessive retinopathy**	**4**	**3**
*NR2F1*	5q15	Dominant optic atrophy with intellectual disability and developmental delay	6	3
*NRL*	14q11.2	Dominant retinitis pigmentosa; recessive retinitis pigmentosa	6	6
*NYX*	Xp11.4	X-linked congenital stationary night blindness	3	2
*OAT*	10q26.13	Recessive gyrate atrophy	8	2
*OFD1*	Xp22.2	Jobert syndrome; orofaciodigital syndrome 1, Simpson-Golabi-Behmel syndrome 2; X-linked retinitis pigmentosa, severe	9	4
***OPA1***	**3q29**	**Dominant optic atrophy, Kjer type; dominant optic atrophy with sensorineural hearing loss**	**32**	**14**
*OPA3*	19q13.32	Recessive optic atrophy with ataxia and 3-methylglutaconic aciduria; dominant optic atrophy with cataract, ataxia and areflexia	3	3
*OPN1LW*	Xq28	Deuteranopia and rare macular dystrophy in blue cone monochromacy with loss of locus control element	3	2
*OPN1MW*	Xq28	Protanopia and rare macular dystrophy in blue cone monochromacy with loss of locus control element	3	2
*OPN1SW*	7q32.1	Dominant tritanopia	1	1
*OTX2*	14q22.3	Dominant Leber congenital amaurosis and pituitary dysfunction; recessive microphthalmia; dominant pattern dystrophy	11	11
*PANK2*	20p13	Recessive HARP (hypoprebetalipoproteinemia, acanthocytosis, retinitis pigmentosa, and palladial degeneration); recessive Hallervorden-Spatz syndrome	12	6
*PAX2*	10q24.31	Dominant renal-coloboma syndrome	9	6
*PCARE*	2p23.2	Recessive retinitis pigmentosa	2	1
*PCDH15*	10q21.1	Recessive Usher syndrome, type 1f; recessive deafness without retinitis pigmentosa; digenic Usher syndrome with CDH23	36	31
*PCYT1A*	3q29	Recessive cone-rod dystrophy with skeletal disease	16	9
*PDE6A*	5q33.1	Recessive retinitis pigmentosa	5	3
*PDE6B*	4p16.3	Recessive retinitis pigmentosa; dominant congenital stationary night blindness	12	8
*PDE6C*	10q23.33	Recessive cone dystrophy, early onset; recessive complete and incomplete achromatopsia	2	1
*PDE6G*	17q25.3	Recessive retinitis pigmentosa	5	3
*PDE6H*	12p12.3	Recessive achromatopsia, incomplete	1	1
*PDZD7*	10q24.31	Recessive non-syndromic deafness	9	7
*PEX1*	7q21.2	Recessive Refsum disease, infantile form	10	3
*PEX2*	8q21.13	Recessive Refsum disease, infantile form	6	5
*PEX7*	6q23.3	Recessive Refsum disease, adult form	3	3
*PGK1*	Xq21.1	Retinitis pigmentosa with myopathy	6	2
*PHYH*	10p13	Recessive Refsum disease, adult form	7	5
*PITPNM3*	17p13.2	Dominant cone-rod dystrophy	5	3
*PLA2G5*	1p36.13-p36.12	Recessive benign fleck retina	8	1
*PLK4*	4q28.2	Recessive microcephaly, growth failure and retinopathy	11	6
*PNPLA6*	19p13.2	Recessive Boucher-Neuhauser syndrome with chorioretinal dystrophy	25	14
*POC1B*	12q21.33	Recessive cone-rod dystrophy; recessive Joubert syndrome	14	4
*POC5*	5q13.3	Recessive syndromic disease with retinitis pigmentosa	14	7
*POMGNT1*	1p34.1	Recessive retinitis pigmentosa	10	3
*PRCD*	17q25.1	Recessive retinitis pigmentosa	13	2
*PRDM13*	6q16.2	Dominant macular dystrophy, North Carolina type; dominant progressive bifocal chorioretinal atrophy	2	1
*PROM1*	4p15.32	Recessive retinitis pigmentosa with macular degeneration; dominant Stargardt-like macular dystrophy; dominant macular dystrophy, bull’s-eye; dominant cone-rod dystrophy	22	11
***PRPF3***	**1q21.2**	**Dominant retinitis pigmentosa**	**7**	**1**
***PRPF31***	**19q13.42**	**Dominant retinitis pigmentosa**	**9**	**6**
***PRPF4***	**9q32**	**Dominant retinitis pigmentosa**	**3**	**2**
***PRPF6***	**20q13.33**	**Dominant retinits pigmentosa**	**1**	**1**
***PRPF8***	**17p13.3**	**Dominant retinitis pigmentosa**	**16**	**5**
*PRPH2*	6p21.1	Dominant retinitis pigmentosa; dominant macular dystrophy; digenic RP with ROM1; dominant adult vitelliform macular dystrophy; dominant cone-rod dystrophy; dominant central areolar choroidal dystrophy; recessive LCA	1	1
*PRPS1*	Xq22.3	Neuropathy, optic atrophy, deafness and retinitis pigmentosa	25	10
*RAB28*	4p15.33	Recessive cone-rod dystrophy	8	7
*RAX2*	19p13.3	Cone-rod dystrophy, isolated; age-related macular degeneration, isolated	2	2
*RB1*	13q14.2	Dominant germline or somatic retinoblastoma; benign retinoma; pinealoma; osteogenic sarcoma	9	4
*RBP3*	10q11.22	Recessive retinitis pigmentosa	1	1
*RBP4*	10q23.33	Recessive RPE degeneration	4	4
*RCBTB1*	13q14.2	Recessive syndromic and non-syndromic retinal dystrophy; dominant familial exudative vitreoretinopathy and Coats disease	4	2
*RD3*	1q32.3	Recessive Leber congenital amaurosis	2	1
*RDH11*	14q24.1	Recessive retinitis pigmentosa, syndromicxd	11	6
*RDH12*	14q24.1	Recessive Leber congenital amaurosis with severe childhood retinal dystrophy; dominant retinitis pigmentosaxd	4	2
*RDH5*	12q13.2	Recessive fundus albipunctatus; recessive cone dystrophy, late onset	11	4
*REEP6*	19p13.3	Recessive retinitis pigmentosa	4	2
*RGR*	10q23.1	Recessive retinitis pigmentosa; dominant choroidal sclerosis	18	9
*RGS9*	17q24.1	Recessive delayed cone adaptation	12	4
*RGS9BP*	19q13.12	Recessive delayed cone adaptation	1	1
***RHO***	**3q22.1**	**Dominant retinitis pigmentosa; dominant congenital stationary night blindness; recessive retinitis pigmentosa**	**1**	**1**
*RIMS1*	6q13	Dominant cone-rod dystrophy	20	16
*RLBP1*	15q26.1	Recessive retinitis pigmentosa; recessive Bothnia dystrophy; recessive retinitis punctata albescens; recessive Newfoundland rod-cone dystrophy	4	2
*ROM1*	11q12.3	Dominant retinitis pigmentosa; digenic retinitis pigmentosa with PRPH2	5	4
*RP1*	8q12.1	Dominant retinitis pigmentosa; recessive retinitis pigmentosa	8	3
*RP1L1*	8p23.1	Dominant occult macular dystrophy; recessive retinitis pigmentosa	2	1
*RP2*	Xp11.23	X-linked retinitis pigmentosa; X-linked retinitis pigmentosa, dominant	1	1
*RP9*	7p14.3	Dominant retinitis pigmentosa	4	2
*RPE65*	1p31.2	Recessive Leber congenital amaurosis; recessive retinitis pigmentosa; dominant retinitis pigmentosa with choroidal involvement	1	1
***RPGR***	**Xp11.4**	**X-linked retinitis pigmentosa, recessive; X-linked retinitis pigmentosa, dominant; X-linked cone dystrophy 1; X-linked atrophic macular dystrophy, recessive**	**10**	**9**
*RPGRIP1*	14q11.2	Recessive Leber congenital amaurosis; recessive cone-rod dystrophy	13	8
*RPGRIP1L*	16q12.2	Recessive Joubert syndrome; recesssive Meckel syndrome	12	10
*RS1*	Xp22.13	Retinoschisis	2	1
*RTN4IP1*	6q21	Recessive optic atrophy, non-syndromic and syndromic	4	2
*SAG*	2q37.1	Recessive Oguchi disease; recessive retinitis pigmentosa; dominant retinitis pigmentosa	16	3
*SAMD11*	1p36.33	Recessive retinitis pigmentosa	17	13
*SDCCAG8*	1q43	Recessive nephronophthisis, ciliopathy-related; recessive Bardet-Biedl syndrome	11	3
*SEMA4A*	1q22	Dominant retinitis pigmentosa; dominant cone-rod dystrophy	16	9
*SLC24A1*	15q22.31	Recessive congenital stationary night blindness	12	7
*SLC25A46*	5q22.1	Recessive syndromic optic atrophy; protein	8	5
*SLC38A8*	16q23.2-q24.2	Recessive foveal hypoplasia and anterior segment dysgenesis	4	3
*SLC7A14*	3q26.2	Recessive retinitis pigmentosa	2	1
***SNRNP200***	**2q11.2**	**Dominant retinitis pigmentosa**	**9**	**2**
*SPATA7*	14q31.3	Recessive Leber congenital amaurosis; recessive RP, juvenile	22	8
*SPP2*	2q37.1	Dominant retinitis pigmentosa	4	3
*TEAD1*	11p15.3	Dominant atrophia areata	6	5
*TIMM8A*	Xq22.1	Optic atrophy with deafness-dystonia syndrome	4	2
*TIMP3*	22q12.3	Dominant Sorsby’s fundus dystrophy	1	1
*TLR3*	4q35.1	Age-related macular degeneration, complex etiology	5	3
*TLR4*	9q33.1	Age-related macular degeneration, complex etiology	4	3
*TMEM126A*	11q14.1	Recessive non-syndromic optic atrophy	6	3
*TMEM216*	11q12.2	Recessive Joubert syndrome; recessive Meckel syndrome	5	2
*TMEM237*	2q33.1	Recessive Jobert syndrome	13	3
*TOPORS*	9p21.1	Dominant retinitis pigmentosa	2	2
*TREX1*	3p21.31	Dominant retinal vasculopathy with cerebral leukodystrophy; dominant Aicardi-Goutiere syndrome 1, dominant chilblain lupus	6	6
*TRIM32*	9q33.1	Recessive Bardet-Biedl syndrome; recessive limb-girdle muscular dystrophy	3	3
*TRNT1*	3p26.2	Recessive retinitis pigmentosa with erythrocytic microcytosis; recessive retinitis pigmentosa, non-syndromic	20	7
*TRPM1*	15q13.3	Recessive congenital stationary night blindness, complete	12	7
*TSPAN12*	7q31.31	Dominant familial exudative vitreoretinopathy	7	6
*TTC8*	14q32.11	Recessive Bardet-Biedl syndrome; recessive retinitis pigmentosa	13	7
*TTLL5*	14q24.3	Recessive cone and cone-rod dystrophy	22	7
*TTPA*	8q12.3	Recessive retinitis pigmentosa and/or recessive or dominant ataxia	2	1
*TUB*	11p15.4	Recessive retinal dystrophy and obesity	3	3
*TUBGCP4*	15q15.3	Recessive chorioretinopathy and microcephaly	12	4
*TUBGCP6*	22q13.33	Recessive microcephaly with chorioretinopathy	8	4
*TULP1*	6p21.31	Recessive retinitis pigmentosa; recessive Leber congenital amaurosis	8	4
*UNC119*	17q11.2	Dominant cone-rod dystrophy	8	6
*USH1C*	11p15.1	Recessive Usher syndrome, Acadian; recessive deafness without retinitis pigmentosa	11	5
*USH1G*	17q25.1	Recessive Usher syndrome	2	1
***USH2A***	**1q41**	**Recessive Usher syndrome, type 2a; recessive retinitis pigmentosa**	**5**	**2**
*VCAN*	5q14.3	Dominant Wagner disease and erosive vitreoretinopathy	12	7
*WDPCP*	2p15	Recessive Bardet-Biedl syndrome	17	7
*WDR19*	4p14	Recessive renal, skeletal and retinal anomalies; recessive Senior-Loken syndrome	18	4
*WFS1*	4p16.1	Recessive Wolfram syndrome; dominant low frequency sensorineural hearing loss	9	6
*WHRN*	9q32	Recessive Usher syndrome, type 2; recessive deafness without retinitis pigmentosa	8	5
*ZNF408*	11p11.2	Dominant familial exudative vitreoretinopathy; recessive retinitis pigmentosa with vitreal alterations	5	1
*ZNF423*	16q12.1	Recessive Jobert syndrome; recessive nephronophthisis	8	8
*ZNF513*	2p23.3	Recessive retinitis pigmentosa	4	3

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
