# Peer review of "The Alter Retina: Alternative Splicing of Retinal Genes in Health and Disease"

_ijms, 2021, doi:10.3390/ijms22041855_

Round 1
Reviewer 1 Report
This is a well written review manuscript, which has comprehensively summarized the roles of alternative splicing in retinal diseases. The references are updated and have included a number of recently published studies. Table 1 has a long list of IRDs with splicing variants. Discussions on the basic biology of alternative splicing and the applications in retinal diseases are informative. I have only a few minor suggestions.
- Line 76, “splicing mutations account for 1/3 of disease-causing mutations”. The statement is not accurate. In the cited reference 15, it was stated that “about 1/3 of disease-causing mutations may disrupt mRNA splicing”. This sentence needs to be revised, similar to Line 304, or further elaborated with additional references.
- Line 555, siRNA-based approaches in treating patients with age-related macular degeneration can have unusual complications (PMID: 21988875; 32708811). Please revise the statement.
- The authors may consider adding the paper on CWC27 and IRD (PMID: 28285769)
Author Response
Reviewer #1:
This is a well written review manuscript, which has comprehensively summarized the roles of alternative splicing in retinal diseases. The references are updated and have included a number of recently published studies. Table 1 has a long list of IRDs with splicing variants. Discussions on the basic biology of alternative splicing and the applications in retinal diseases are informative. I have only a few minor suggestions.
Line 76, “splicing mutations account for 1/3 of disease-causing mutations”. The statement is not accurate. In the cited reference 15, it was stated that “about 1/3 of disease-causing mutations may disrupt mRNA splicing”. This sentence needs to be revised, similar to Line 304, or further elaborated with additional references.
We thank the reviewer for its recommendations to improve the manuscript. We agree with the reviewer that this sentence is not accurate. We have revised the bibliography and modified our sentences in the text (see lines 76–78 and 316). We have now referred to the percentage of splicing mutations reported in the HGMD.
Line 555, siRNA-based approaches in treating patients with age- related macular degeneration can have unusual complications (PMID: 21988875; 32708811). Please revise the statement.
In agreement with the reviewer, we have added more information regarding siRNA therapies, including references to the side effects associated to siRNA administration (see lines 577–581).
The authors may consider adding the paper on CWC27 and IRD (PMID: 28285769)
We thank the reviewer for suggesting the addition of the CWC27 paper. We agree that it may be a good example to complement our manuscript. We have modified our sentences in the text to include the CWC27 example (see lines 247–249 and 355) and introduced two references. Besides, we have checked the shift in the numbering of references within the text due to the addition of these articles.
Reviewer 2 Report
The review 'The alter retina: alternative splicing of retinal genes in health 2
and disease' by Aisa-Marin et al. explores alternative splicing and its implications on retinal health and disease. The review is well written and contains a wealth of information on how splicing works and what might go wrong in disease (with a focus on the retina).
I think the review can be published in its current form. There are just a few grammatical errors that should be dealt with:
There are a few double spaces after fullstops; e.g. l 64.
l 15f - The retina expresses retinal specific splicing factors and produces a large number of alternative transcripts
l 61f - the central nervous system (CNS) of vertebrates (CNS)
l 75f - Usher syndrome (USH)
l 103f - Despite Even though they represent only 1% of all AS
l 107 - are associated to with late (this has to be changed throughout the text)
line 141 - critical roles in the retina development
line 170 - which does not
line 217 - sequences around (or surrounding) the adenosine
l 268f - Besides, RBFOX1 and RBFOX2 may be important
l 315 - proteinc isoforms
l 340 - ActuallyIn fact, they represent
l 382 - Usher syndrome (USH)
l 455 - Usher syndrome (USH)
l544ff - This sentence is too long. I would do something like this: In this context, it is worth noting the use of siRNA and shRNA agents are worth noting. SiRNA has shown potential in patients with age-related macular dystrophy (AMD) [124], and shRNA is particularly beneficial in the treatment of autosomal dominant disorders, such as those caused by RHO mutations [125] as well as in silencing VEGF production in AMD mouse models [126].
l 572f - differential different
l 578 - ActuallyIn fact, identification
Author Response
Reviewer #2:
The review 'The alter retina: alternative splicing of retinal genes in health and disease' by Aisa-Marin et al. explores alternative splicing and its implications on retinal health and disease. The review is well written and contains a wealth of information on how splicing works and what might go wrong in disease (with a focus on the retina).
I think the review can be published in its current form. There are just a few grammatical errors that should be dealt with:
We thank the reviewer for its comments and apologize for the grammatical errors. We have carefully checked the grammar throughout the text and accordingly modified our sentences. There is a full list below of all modifications and corrections suggested by the reviewer as well as our amendments.
There are a few double spaces after fullstops; e.g. l 64.
We have checked all the double spaces throughout the text and corrected them.
l 15f - The retina expresses retinal specific splicing factors and produces a large number of alternative transcripts. Corrected, l 16
l 61f - the central nervous system (CNS) of vertebrates (CNS). Corrected, l 62
l 75f - Usher syndrome (USH). Changed, l 76
l 103f - Despite Even though they represent only 1% of all AS. Changed, l 107
l 107 - are associated to with late (this has to be changed throughout the text). Corrected, l 112
l 141 - critical roles in the retina development. Corrected, l 145
l 170 - which does not. Corrected, l 178
l 217 - sequences around (or surrounding) the adenosine. Corrected, l 225
l 268f - Besides, RBFOX1 and RBFOX2 may be important. Deleted, l 280
l 315 - proteinc isoforms. Corrected, l 327
l 340 - ActuallyIn fact, they represent. Changed, l 356
l 382 - Usher syndrome (USH). Changed, l 398
l 455 - Usher syndrome (USH). Changed, l 476
l544ff - This sentence is too long. I would do something like this: In this context, it is worth noting the use of siRNA and shRNA agents are worth noting. SiRNA has shown potential in patients with age-related macular dystrophy (AMD) [124], and shRNA is particularly beneficial in the treatment of autosomal dominant disorders, such as those caused by RHO mutations [125] as well as in silencing VEGF production in AMD mouse models [126].
We have divided this long sentence into several shorter ones. As Reviewer 1 also suggested some changes to this particular sentence, the new version addresses the concerns of the two reviewers. See lines 577-584.
l 572f - differential different. Corrected, l 598-599
l 578 - ActuallyIn fact, identification. Changed, l 604